# Norovirus Seroprevalence among Adults in the United States: Analysis of NHANES Serum Specimens from 1999–2000 and 2003–2004

**DOI:** 10.3390/v12020179

**Published:** 2020-02-05

**Authors:** Amy E. Kirby, Yvonne Kienast, Wanzhe Zhu, Jerusha Barton, Emeli Anderson, Melissa Sizemore, Jan Vinje, Christine L. Moe

**Affiliations:** 1Rollins School of Public Health, Emory University, Atlanta, GA 30322, USA; yvonne.kienast@emory.edu (Y.K.); wanzhez@gmail.com (W.Z.); jerusha.barton@dph.ga.gov (J.B.); emeli.anderson@emory.edu (E.A.); melissa.anne.sizemore@alumni.emory.edu (M.S.); clmoe@emory.edu (C.L.M.); 2Division of Viral Diseases, Centers for Disease Control and Prevention, Atlanta, GA 30329, USA; ahx8@cdc.gov

**Keywords:** norovirus, seroprevalence, Norwalk virus, NHANES

## Abstract

Norovirus is the most common cause of epidemic and endemic acute gastroenteritis. However, national estimates of the infection burden are challenging. This study used a nationally representative serum bank to estimate the seroprevalence to five norovirus genotypes including three GII variants: GI.1 Norwalk, GI.4, GII.3, GII.4 US95/96, GII.4 Farmington Hills, GII.4 New Orleans, and GIV.1 in the USA population (aged 16 to 49 years). Changes in seroprevalence to the three norovirus GII.4 variants between 1999 and 2000, as well as 2003 and 2004, were measured to examine the role of population immunity in the emergence of pandemic GII.4 noroviruses. The overall population-adjusted seroprevalence to any norovirus was 90.0% (1999 to 2000) and 95.9% (2003 to 2004). Seroprevalence was highest to GI.1 Norwalk, GII.3, and the three GII.4 noroviruses. Seroprevalence to GII.4 Farmington Hills increased significantly between the 1999 and 2000, as well as the 2003 and 2004, study cycles, consistent with the emergence of this pandemic strain. Seroprevalence to GII.4 New Orleans also increased over time, but to a lesser degree. Antibodies against the GIV.1 norovirus were consistently detected (population-adjusted seroprevalence 19.1% to 25.9%), with rates increasing with age. This study confirms the high burden of norovirus infection in US adults, with most adults having multiple norovirus infections over their lifetime.

## 1. Introduction

Norovirus is the most common cause of acute gastroenteritis (AGE) in the United States and accounts for an estimated 58% of foodborne AGE [1,2]. The disease is characterized by an abrupt onset of diarrhea and/or vomiting, after a short incubation period (12 to 48 h) [3,4]. Generally, symptoms resolve within one to three days without intervention. However, severe dehydration can occur in 10% to 12% of individuals, which can lead to hospitalization and, in rare cases, death. There is no specific treatment for norovirus gastroenteritis; cases are treated with rehydration therapy and other supportive care. People of all ages are susceptible to norovirus infection, although severe outcomes are more likely in the very young (< 2 years) and the elderly (> 70 years) [4,5]. The only host factor known to correlate with susceptibility to infection are histo-blood group antigens (HBGAs), which are carbohydrates that serve as binding ligands for virus entry into the target host cells [6,7]. HBGA production is encoded by gene families expressing the ABO (A/B enzymes), secretor (α [1,2]-fucosyltransferase 2, or *FUT2*), and Lewis-type (*FUT3*) antigens. Individuals with single-nucleotide polymorphisms in their *FUT2* gene are termed secretor-negative and human challenge studies have shown that these individuals are resistant to infection with Norwalk virus, the prototype genogroup I norovirus [7]. This finding is supported by outbreak investigations that found only secretor-positive individuals were infected [8,9,10]. However, this mutation does not provide absolute protection as secretor-negative individuals can also be infected with certain norovirus genotypes [11,12,13,14]. The diversity of norovirus genotypes that exhibit this secretor-independent infection, as well as the extent of susceptibility in secretor-negative people, is unknown. In the USA, approximately 75% of the population is secretor-positive [15,16].

Noroviruses are divided into 10 genogroups (G) based on amino acid identity in the capsid protein VP1, but the majority of human infections are caused by viruses from two genogroups, GI and GII [17]. Within the genogroups, viruses are further divided into genotypes (GI = 9 and GII = 27 of which GII.11, GII.18, and GII.19 have only been detected in swine). Meta-analysis of published outbreaks have found that waterborne outbreaks are more likely to be associated with GI viruses, whereas GII are more likely associated with person-to-person transmission, although this distinction is not absolute [18]. Over the last 17 years, GII.4 noroviruses have been responsible for at least half of the norovirus outbreaks in the USA and exhibit a pandemic cycle with novel strains emerging every few years [19]. Data suggest that the GII.4 pandemic cycle is likely driven by herd immunity which selects for emerging pandemic strains with different antigenic features from the currently predominant strain [20,21]. Although GIV infections are rare and very little is known about the prevalence of these viruses, [22] analysis of municipal sewage suggests that these viruses do circulate in humans [23,24,25].

Estimating the burden of norovirus infection in the USA is challenging [26]. The high diversity of noroviruses [27], rapid evolution of novel strains [28,29], and relatively short-lived acquired immunity [30,31] mean that most individuals will be infected many times over their lifetime. Due to the generally mild nature and short duration of the disease, most people do not seek medical treatment. Of those who do, most will not have a stool sample tested for norovirus, and there is no requirement to report individual cases. Thus, the documented cases are primarily those associated with large outbreaks or severe outcomes. In addition, approximately 30% of infections are asymptomatic [7,32]. 

Researchers have used a variety of approaches to overcome the challenges of estimating the burden of norovirus infection. Outbreak reporting [2], hospital admissions [33], deaths [34], and secondary analysis of hospital stool samples [35] have been used to estimate norovirus burden. Each of these approaches found a high infection burden, with a generally accepted estimate of 21 million infections annually in the USA. Despite the high burden estimates, these are likely underestimates due to the lack of data on sporadic and asymptomatic cases. 

An alternative approach is to determine the prevalence of anti-norovirus antibodies as a surrogate for infection status. This approach has been used to estimate norovirus infection burden in many countries [36,37,38,39,40,41,42,43,44,45,46]. Seroprevalence against norovirus in adults in high-income nations ranged from 59% to 93% and approached 100% among adults in low- and middle-income countries. These studies confirm that norovirus infections occur very early in life, with seroprevalence levels at 12 to 24 months of age ranging from 12% to 97%, depending on the population and the viral strain used as antigen. Although these studies confirm that the burden of norovirus infection is high in all age groups, they are of limited utility in estimating the actual disease burden in the general population. Although seroprevalence is a more sensitive measure of population-level norovirus infection burden, it is not an estimate of disease burden because it cannot distinguish between symptomatic and asymptomatic infection. 

In this study, archived serum specimens from the 1999–2000 and 2003–2004 National Health and Nutrition Examination Survey (NHANES, [47]) study cycles were analyzed for IgG antibodies against a panel of seven norovirus virus-like particles (VLPs). VLPs are composed of capsid proteins that are antigenically and morphologically indistinguishable from live norovirus particles, but they do not contain any genomic RNA [48]. The NHANES participants were selected based on census data to be statistically representative of the healthy, non-institutionalized USA population in the year of the survey. The following VLPs were chosen to represent the diversity of genotypes for each genogroup: GI.1 Norwalk, GI.4, GII.3, and GIV.1. In addition, three GII.4 variants (US95/96, Farmington Hills 2002, and New Orleans 2009) were included to examine the immune evasion hypothesis of GII.4 pandemic emergence because GII.4 US95/96 emerged before the first study cycle, GII.4 Farmington Hills emerged during the study period, and GII.4 New Orleans emerged after the second study cycle. This study provides a nationally representative estimate of norovirus seroprevalence in the USA.

## 2. Materials and Methods 

### 2.1. Serum Specimens 

Serum specimens were acquired from the National Health and Nutrition Examination Survey (NHANES) biorepository [47]. The study was approved by the Centers for Disease Control and Prevention Institutional Review Board (CDC protocol ID 2013-10). Specimens were selected from the 1999 to 2000 and 2003 to 2004 study cycles, which were before and after, respectively, the emergence of the pandemic GII.4 Farmington Hills norovirus strain in 2002 [49]. A one-third subset of specimens from subjects between the ages of 16 and 49 was selected from each study cycle. This age range comprised the bulk of the population and was found to have the most social contacts with the very young and the very old in the POLYMOD network analysis [50]. The biorepository does not store specimens from children under 6 years of age and has very limited samples from participants over 65 years of age [47]. The subset was selected such that it would maintain national representativeness with respect to age, gender, and race. 

### 2.2. Norovirus VLP Panel

Norovirus virus-like particles (VLPs) were used as antigens. The VLPs were generously provided by Dr. Robert Atmar and the Baylor VLP Production Facility. The VLPs were selected based on the strains known to be circulating in the USA during the two NHANES study cycles. The following VLPs were included in the panel: GI.1 Norwalk, GI.4, GII.3, GII.4 US 95/96, GII.4 Farmington Hills, GII.4 New Orleans, and GIV.1. The three GII.4 strains emerged before (GII.4 US 95/96, first detected in 1995 [51,52]), during (GII.4 Farmington Hills, detected in 2002 [49]), and after (GII.4 New Orleans, detected in 2009 [53]) the two NHANES study cycles included in this analysis. 

### 2.3. Anti-Norovirus IgG ELISA

Direct enzyme-linked immunosorbent assays (ELISAs) were used to determine anti-norovirus serum IgG titers. Medium binding polystyrene plates were coated with 2 µg/mL norovirus VLPs. Unbound VLPs were removed by washing the plate 5 times in PBS-T (phosphate buffered saline + 0.025% Tween 20). The plates were blocked overnight at 4 °C with 5% Blotto in PBS-T. After 5 washes, the plates were incubated for 1 h at room temperature with duplicate serum samples diluted 1:50 in blocking solution. Unbound antibody was removed by washing 5 times with PBS-T. The secondary antibody (alkaline phosphatase-labeled rabbit α-human IgG, Sigma-Aldrich Co., St. Louis, MO, USA) was diluted 1:2500 in blocking solution and incubated for 30 min at room temperature. After a final 5 washes in PBS-T, 100 µL of *p*-nitrophenyl phosphate solution (Sigma-Aldrich Co.) was added to each well, and the plate was incubated at room temperature in the dark for 10 to 30 min. The optical density (OD) at 405 nm was determined using an ELx800 plate reader (BioTek Instruments, Inc. Winooski, VT, USA). For each norovirus VLP evaluated, positive control serum from confirmed cases was included on each plate, except for GIV.1, where positive serum from a previous seroprevalence study was used. 

### 2.4. Determination of OD Cut-Points

The OD cut-points for analysis were selected based on serology results from a previous norovirus human challenge study [54]. The study challenged healthy adults (age range 18 to 48 years) with GI.1 Norwalk, a strain that had not circulated widely in the USA population for decades. Thus, the serological response to the challenge was a reasonable measure of the immune response in adults where any pre-existing reactivity was assumed to be due to cross-reactivity with previous infections. Similarly, the immune response 30 days post-challenge represented the immune response to a recent homologous infection. The distribution of serological responses in the subjects was used to determine the following two cut-points: OD ≥ 1.5, based on the serological distribution in uninfected individuals, defines seropositivity; and OD ≥ 3.0, based on the serological distribution of infected individuals, is evidence of recent infection. The misclassification frequency for each cut-point was calculated. At the low cut-point (OD ≥ 1.5), 100% of infected subjects were correctly classified as seropositive and 54% of uninfected subjects were correctly classified as seronegative. At the high cut-point (OD ≥ 3.0) used to identify recent infections, 87% of infected subjects were correctly classified as seropositive and 74% of uninfected subjects are correctly classified as seronegative. Due to the large variation in immune response, there were no cut-points that correctly classified all subjects. 

### 2.5. Statistical Analysis

SAS version 9.4 (SAS Institute Inc., Cary, NC, USA) was used for all statistical analyses. Mean OD values for each specimen-VLP pair were evaluated based on two cut-points. Mean OD values greater than or equal to 1.5 were considered seropositive. Mean OD values greater than or equal to 3.0 were considered highly seropositive and evidence of a recent infection. Seropositivity due to infection could not be distinguished from cross-reactivity. There was no correction for potential cross-reactivity. Age, reported in years at screening, was stratified into the following four groups: 16 to 19 years, 20 to 29 years, 30 to 39 years, and 40 to 49 years. Race was categorized into five groups based on NHANES data as follows: non-Hispanic white, non-Hispanic black, Mexican American, other Hispanic, and other (which included multiracial). The seroprevalence to each VLP, in each survey cycle, was calculated overall and by age, gender, and race. Wald 95% confidence intervals were computed for all proportions, and differences were assessed using two-sided Student’s *t*-test at the 0.05 significance level. Weights provided by NHANES were used to account for the complex sample design. The weights were multiplied by three to adjust for the one-third subset. 

### 2.6. Data Availability

Serum antibody concentration data are available for download from the NHANES website [47]. 

## 3. Results

Serum specimens from the 1999 to 2000 (*N* = 1051) and 2003 to 2004 (*N* = 1102) survey cycles were analyzed for antibodies to a panel of seven norovirus VLPs. The effective sample size was 128,030,088 after adjusting for the sample weights. The overall population-adjusted norovirus seroprevalence was 90.0% and 95.9% in the 1999 to 2000 and 2003 to 2004 study cycles, respectively (Table 1). The norovirus seroprevalence for any recent infection was 51.8% and 59.1%, respectively. Among all seven antigens seropositivity was rare (1.0% and 3.0%, respectively) and no samples had evidence of recent infection from all seven norovirus strains. Between the two study cycles, there was a statistically significant decrease in the proportion of completely seronegative subjects, from 10.0% to 4.1% (*p* < 0.001). 

Overall norovirus seroprevalence, defined as an OD ≥ 1.5 for at least one antigen, did not vary significantly among age groups (Figure 1A, ANOVA *p* = 0.9460). However, there was a statistically significant difference in the age distribution of the seropositive samples between the two study cycles (ANOVA *p* < 0.001). This difference was driven by increases in seroprevalence in the 16 to 19 and 30 to 39 age groups. Approximately half of subjects in all age groups had evidence of recent infection with at least one norovirus strain, with the exception of the 16 to 19 age group in the 2003 to 2004 study cycle (Figure 1B). In the 2003 to 2004 study cycle, the 16 to 19 age group was more likely to have evidence of recent infection than all other age groups in that study cycle (*p* = 0.004), as well as the same age group in the 1999 to 2000 study cycle (*p* = 0.0003).

Seroprevalence varied considerably by antigen (Table 2). Roughly half of the subjects were seropositive for GI.1 Norwalk, GII.3, and GII.4 US95/96 in both study cycles. Seroprevalence to GII.4 New Orleans antibodies was slightly lower (41.6% and 48.1%). Seroprevalance to GI.4 and GIV.1 was the lowest, with 27.6% or less of the sample set meeting the cut-point. GII.4 Farmington Hills is the only antigen with a statistically significant change in seroprevalence between the study cycles, increasing from 36.9% to 57.5% (*p* < 0.001). Using the higher cut-point to define recent infection yielded similar trends in seroprevalence by antigen, although the proportions were much lower. In both study cycles, seroprevalence was highest to GII.3 and GII.4 US95/96, with proportions ranging from 26.1% to 29.9%. High reactivity (OD ≥ 3.0) to GI.4, GII.4 New Orleans, and GIV.1 antigen was rare (≤ 6.4%). Between the two study cycles, the proportion of samples with high reactivity to GII.4 Farmington Hills increased from 4.5% to 11.7% (*p* < 0.001).

In the 2003 to 2004 study cycles, in all age groups, the GII.4 Farmington Hills seropositivity at both cut-points was significantly higher (Figure 2, *p* < 0.05 for both). Among the age groups, there was no significant difference in seropositivity at either cut-point. GI.4, GII.3, and GII.4 US 95/96 also had an increase in seropositivity between the 1999 to 2000 and 2003 to 2004 study cycles, but only in the 16 to 19 age group (Appendix A). Gender and race were not associated with seroprevalence (data not shown).

## 4. Discussion

Overall seroprevalence to any of the tested norovirus antigens was high (≥ 90.0%). However, there was significant variation among genotypes, age groups, and study cycles. The increase in seroprevalence from the 1999–2000 study cycle to the 2003–2004 study cycle is consistent with the increased incidence associated with norovirus epidemic years [51,55] and the emergence of GII.4 Farmington Hills in 2002 [49]. Indeed, seropositivity to GII.4 Farmington Hills increased from 36.9% to 57.5% for overall infections and from 4.5% to 11.7% for recent infections. Modest, but statistically significant increases in seroprevalence to GII.4 New Orleans and GIV.1 also contributed to the overall increase in norovirus seroprevalence. A small proportion of the population (10.0%) was seronegative for all seven antigens. This can be partially explained by the decreased susceptibility of secretor-negative individuals, who make up approximately 25% of the USA population [7,15,16,56]. Not all norovirus genotypes are secretor-dependent [11,12], which could explain why the proportion of pan-seronegatives is lower than the proportion of secretor-negatives. 

The seroprevalence to GI.1 Norwalk was higher than expected from epidemiologic data. GI infections are diagnosed less frequently than GII infections [19]. However, the seroprevalence data aligns well with sewage studies that report GI virus concentrations and frequencies at levels similar to GII viruses [57]. This finding suggests that GI infections are under-diagnosed relative to GII infections. In human challenge studies, GI infections cause less severe illness than GII infections, which could lead to fewer GI cases seeking medical care [3]. GII infections are also more likely to be associated with person-to-person spread and large outbreaks, which increases the likelihood of a lab-confirmed diagnosis [18]. 

GIV noroviruses are not routinely tested for, thus, the available prevalence data is scarce. Similar to GI viruses, GIV noroviruses are frequently detected in sewage studies, suggesting that they do have a stable circulation within human populations [23,58,59]. The overall GIV seroprevalence reported in this study is similar to that reported in a hospital-based study of Italian children and adults, which reported an overall GIV seroprevalence of 28.2% [60]. Both studies reported a significant increase in seroprevalence with increasing age. 

Epitope mapping studies have suggested that the emergence of new GII.4 pandemic strains is driven by an overall immunity within the population [21,30]. If this is correct, a population should have high seroprevalence rates to current and past epidemic strains, but lower reactivity to future pandemic strains. The results of this study do not clearly support or refute that hypothesis. In the 1999–2000 cohort, seroprevalence was highest to GII.4 US95/96, the dominant strain at the time, and lower to GII.4 Farmington Hills and GII.4 New Orleans, which had not yet emerged (Table 2). The difference was even more pronounced at the higher OD cut-point. However, in the 2003 to 2004 cohort, the seroprevalence to both the newly emerged GII.4 Farmington Hills and the future strain GII. 4 New Orleans increased. This concurrent increase could indicate cross-reactivity between the antigens, shared epitopes, or that the strain was circulating in the population earlier than detected. Analysis of carbohydrate-blocking antibodies could provide a clearer picture, as carbohydrate-blocking antibodies are correlated with protection from infection, while total anti-norovirus antibodies are not [61].

A limitation of all seroprevalence studies is selecting an appropriate cut-point to define seropositivity. This is even more challenging when there are many circulating strains with largely unknown cross-reactivity, multiple infections over a lifetime are likely, and the study subjects are adults. In this situation, setting a low cut-point results in all, or nearly all, samples classified as seropositive, as reported in other norovirus seroprevalence studies [36,37,39,40,42,43,44,45,46]. To better understand more recent infections and reduce the impact of cross-reactivity, higher cut-points based norovirus human challenge data, were used in this study. This study was also limited by the availability of serum samples from the NHANES biorepository [47]. Samples from children and the elderly are either unavailable or very limited in number. Additionally, NHANES excludes institutionalized individuals, which excludes populations in hospitals and long-term care facilities where most norovirus outbreaks are reported [1]. 

## 5. Conclusions

Using a nationally representative serum collection, this study confirms that norovirus is a common infection in the USA with most adults having multiple infections over their lifetime. These data indicate that GI.1 Norwalk, GII.3, and GII.4 viruses are the most common noroviruses circulating in the USA during the study cycle periods. Antibodies against the GIV.1 norovirus were consistently detected, in all age groups, suggesting that this virus circulates in the general population. However, it remains unclear whether these viruses are causing symptomatic disease. This study did not support or refute the role of population immunity in GII.4 pandemic emergence. A norovirus vaccine is needed to reduce the significant infection burden associated with these viruses. 

## Figures and Tables

**Figure 1 viruses-12-00179-f001:**
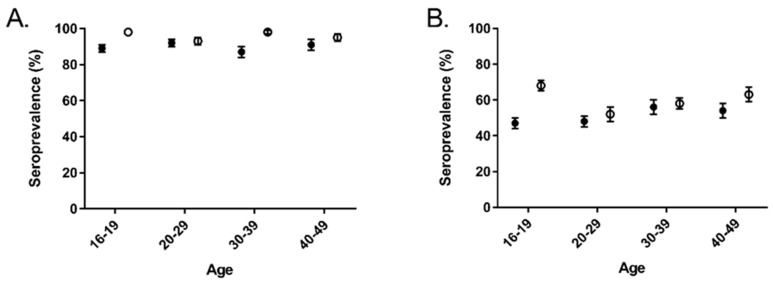
Overall seroprevalence and serologic evidence of recent infection by age and the National Health and Nutrition Examination Survey (NHANES) study cohort. Closed circles, 1999 to 2000 and open circles, 2003 to 2004. (**A**) Proportion seropositive defined as OD ≥ 1.5; (**B**) Proportion with evidence of recent infection defined as OD ≥ 3.0.

**Figure 2 viruses-12-00179-f002:**
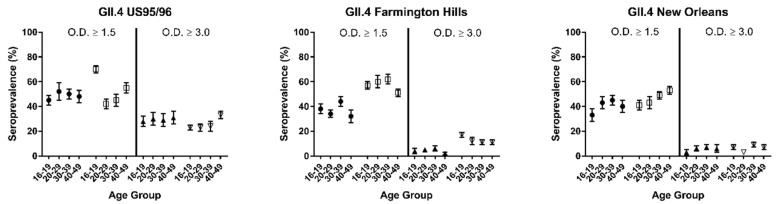
Overall seroprevalence and serologic evidence of recent infection stratified by GII.4 antigen, age group, and NHANES study cycle. Overall seroprevalence defined as OD ≥ 1.5. Evidence of recent infection defined as OD ≥ 3.0. Closed circles, 1999–2000 and open circles, 2003–2004.

**Table 1 viruses-12-00179-t001:** Population-adjusted overall norovirus seroprevalence in USA adults aged 16 to 49 years.

	1999–2000		2003–2004		
	Seroprevalence (%)	95% Confidence Limits	Seroprevalence (%)	95% Confidence Limits	*p*-Value ^a^
Seropositive for at least one antigen					
OD ≥ 1.5	90.0	87.5, 92.5	95.9	94.1, 97.6	0.007
OD ≥ 3.0	51.8	48.4, 55.3	59.1	55.4, 62.9	0.003
Seropositive for all 7 antigens					
OD ≥ 1.5	1.0	4.0, 2.5	3.0	1.5, 3.7	0.120
OD ≥ 3.0	0.0	0.0, 0.0	0.0	0.0, 0.0	-
Seronegative for all 7 antigens					
OD < 1.5	10.0	7.5, 12.5	4.1	2.4, 5.9	< 0.001

^a^ 1999–2000 seroprevalence versus 2003–2004 seroprevalence, assessed using a two-sided Student’s *t*-test.

**Table 2 viruses-12-00179-t002:** Population-adjusted seroprevalence by norovirus antigen and NHANES study cycle.

Antigen	1999–2000 Study Cycle		2003–2004 Study Cycle		*p*-Value ^a^
	Seroprevalence (%)	95% Confidence Limits	Seroprevalence (%)	95% Confidence Limits	
Cut-point OD ≥ 1.5					
GI.1 Norwalk	59.7	53.9, 65.5	56.4	51.4, 61.4	0.360
GI.4	27.6	24.4, 30.9	27.0	22.7, 31.3	0.812
GII.3	53.7	49.1, 58.4	50.0	44.1, 55.9	0.302
GII.4 US 95/96	49.6	45.6, 53.5	49.9	44.1, 55.8	0.927
GII.4 Farmington Hills	36.9	33.6, 40.3	57.5	53.5, 61.5	<0.001
GII.4 New Orleans	41.6	37.0, 46.1	48.1	44.3, 51.9	0.019
GIV.1	19.1	14.9, 23.4	25.9	21.9, 29.9	0.014
Cut-point OD ≥ 3.0					
GI.1 Norwalk	15.8	12.3, 19.2	19.5	15.8, 23.3	0.126
GI.4	3.4	1.2, 5.6	6.0	3.7, 8.3	0.081
GII.3	26.1	22.9, 29.2	28.1	22.4, 33.8	0.523
GII.4 US 95/96	29.9	25.4, 34.4	26.8	23.2, 30.3	0.249
GII.4 Farmington Hills	4.5	2.4, 6.6	11.7	9.5, 13.9	<0.001
GII.4 New Orleans	6.1	3.2, 9.0	6.4	4.1, 8.7	0.866
GIV.1	0.9	0.2, 1.5	2.3	1.0, 3.7	0.040

^a^ 1999–2000 seroprevalence versus 2003–2004 seroprevalence, assessed using a two-sided Student’s *t*-test.

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
