# Peer review of "Norovirus Seroprevalence among Adults in the United States: Analysis of NHANES Serum Specimens from 1999–2000 and 2003–2004"

_viruses, 2020, doi:10.3390/v12020179_

Round 1

Reviewer 1 Report

Kirby et al. have performed a serological analysis of human samples from the NHANES biorepository as a method to characterize the prevalence of norovirus infections within the population.

Although the experiments are conducted well and the rationale for the study is sound, the data are incremental and provide no new robust conclusions about the the disease burden of norovirus infections. The epidemiological findings do not contribute anything new to the field.

From the authors own conclusion of the study, it is clear that Kirby et al. have not identified anything novel.

This reviewer believes that this dataset is unlikely to be useful to field.

Author Response

While our overall findings do not contradict previous estimates of norovirus burden, there are several unique aspects of the study which will be of great value to the field. First, this is study used a census-based nationally representative biospecimen set. This dataset allowed us to estimate the burden of norovirus infection in the US population without any assumptions regarding care-seeking behavior, testing frequency, results reporting, etc. It is noteworthy that this estimate supports previous burden estimates. Unlike other estimates, this study includes seroprevalence data on 5 different norovirus genotypes and 3 GII.4 variants. Finally, this study addresses the immune evasion hypothesis using biospecimens from the general US population, where previous studies have used outbreak specimens or specimens from individuals in healthcare settings.  

Reviewer 2 Report

In “Norovirus Seroprevalence in Adults in the United States: Analysis of NHANES Serum Specimens from 1999-2000 and 2003-2004” Kirby et al leverage the large biorepository of the NHANES to explore seroprevalance and changes of seroprevalance over time to 5 different noroviruses (representing 3 distinct genogroups). The study represents a massive effort to understand norovirus exposure to diverse strains overtime, including a strain that emerged in between the sampling period. Overall, the findings are of interest and well reported. A few minor points should be addressed prior to publication.

1.) In its current format, the figures are of too poor quality and resolution.

2.) It isn’t clear why an O.D. 1.5 and 3.0 were chosen for seropositive and recent infection. This reviewer appreciates the magnitude of the data and difficulty choosing a cut off that balances sensitivity and specificity. However, is there an objective means to arriving at the cut-off? Is there precedence for these numbers or a stratification that becomes apparent when looking at control data? This should be discussed and linked to the points raised about cross-reactivity earlier in the manuscript.

3.) Line 248, “only” is a bit strong of a statement as Reeck et al in a small cohort show a correlation of protection between blocking antibodies, but not total antibodies. Other attributes of the antibodies may be correlated but were not tested.

Author Response

All figures are TIFF files with a resolution of at least 600 dpi, per the journal instructions. A section has been added to the Methods section describing the choice and validation of the cutpoints. We agree with the reviewer. The sentence has been revised.

Reviewer 3 Report

The authors examined NoV seroprevalence in adults in USA. Although there were limitations regarding cutoff points and cross-reactivity, the study was well designed and described. I have several minor comments.

1. A sentence (lines 34-36) could be cited a suitable reference.

2. X-axis values of all figures would be more suitable for "percentage" rather than a rate.             

Author Response

The references for the indicated sentences have been added. The y-axis values for all figures have been revised to percentages.

Reviewer 4 Report

This study investigates the seroprevalence of three genogroups (five genotypes) norovirus (GI.1 and 4; GII.3 and 4; GIV.1) using sera that were acquired from nationally representative serum bank in United States. The authors demonstrate that GI.1 Norwalk, GII.3, and GII.4 viruses are the most common noroviruses circulating in the US during the 1999-2000 and 2003-2004. Furthermore, they report that antibodies to GIV.1 noroviruses were consistently detected in all age groups, suggesting that this virus circulates in the general population.

General comments:
Overall, the work tackles an interesting topic and shed new light on the serological survey for all genogroups of human norovirus (GI, GII, and GIV). The scientific rationale for the investigation is sound and the authors present a few novel and important observations. The manuscript is well organized and carried out. However, some information are missing and should be provided to clarify a few issues, and additional experiments are needed to strengthen the study and further support the authors’ conclusions.
The following points should be addressed:

-The authors should provide a rationale for setting the cutpoint OD to >1.5 or >3.0. Please state this reason in the manuscript.

-Were any sera cross-reactive for same genotype of norovirus? For example, the serum that reacted to GI.1 VLP also reacted to GI.4 VLP?

Author Response

A section has been added to the Methods describing the choice and validation of cutpoints. Yes, the full range of seropositivity was seen, from specimens seronegative for all 7 antigens (10%) to specimens seropositive for all 7 antigens (1%) (Table 1). We did not see any clear evidence of cross-reactivity, for example consistent seropositivity in antigen pairs or subsets. Because we could not distinguish between cross-reactivity and past infection, all seropositive results were included in the analyses. Additional language addressing this issue has been added (line 157-159).